# An Investigation of the Relationship between *Cyniclomyces guttulatus* and Rabbit Diarrhoea

**DOI:** 10.3390/pathogens10070880

**Published:** 2021-07-12

**Authors:** Tuanyuan Shi, Xinlei Yan, Hongchao Sun, Yuan Fu, Lili Hao, Yongxue Zhou, Yan Liu, Wenying Han, Guolian Bao, Xun Suo

**Affiliations:** 1Department of Animal Parasitology, Institute of Animal Husbandry and Veterinary Medicine, Zhejiang Academy of Agricultural Science, Hangzhou 310021, China; shity@zaas.ac.cn (T.S.); sunhongchao@zaas.ac.cn (H.S.); fuy@zaas.ac.cn (Y.F.); zhouyx@zaas.ac.cn (Y.Z.); liuyan88@zaas.ac.cn (Y.L.); 2Department of Food Quality and Safety, Food Science and Engineering College of Inner Mongolia Agricultural University, Hohhot 010018, China; yanxinlei@imau.edu.cn (X.Y.); hanwenying1020@emails.imau.edu.cn (W.H.); 3Department of Veterinary Medicine, College of Life Science & Technology, Southwest Minzu University, Chengdu 610041, China; 22100089@swun.cn; 4National Animal Protozoa Laboratory, Department of Preventive Veterinary Medicine of College of Veterinary Medicine, China Agricultural University, Beijing 100093, China

**Keywords:** rabbit, diarrhoea, *Cyniclomyces guttulatus*, *Eimeria intestinalis*, opportunistic pathogen

## Abstract

*Cyniclomyces guttulatus* is usually recognised as an inhabitant of the gastrointestinal (GI) tract in rabbits. However, large numbers of *C. guttulatus* are often detected in the faeces of diarrhoeic rabbits. The relationship of *C. guttulatus* with rabbit diarrhoea needs to be clearly identified. In this study, a *C. guttulatus* Zhejiang strain was isolated from a New Zealand White rabbit with severe diarrhoea and then inoculated into SPF New Zealand white rabbits alone or co-inoculated with *E**imeria*
*intestinalis**,* another kind of pathogen in rabbits. Our results showed that the optimal culture medium pH and temperature for this yeast were pH 4.5 and 40–42 °C, respectively. The sequence lengths of the 18S and 26S ribosomal DNA fragments were 1559 bp and 632 bp, respectively, and showed 99.8% homology with the 18S ribosomal sequence of the NRRL Y-17561 isolate from dogs and 100% homology with the 26S ribosomal sequence of DPA-CGR1 and CGDPA-GP1 isolates from rabbits and guinea pigs, respectively. In animal experiments, the *C. guttulatus* Zhejiang strain was not pathogenic to healthy rabbits, even when 1 × 10^8^ vegetative cells were used per rabbit. Surprisingly, rabbits inoculated with yeast showed a slightly better body weight gain and higher food intake. However, SPF rabbits co-inoculated with *C. guttulatus* and *E. intestinalis* developed more severe coccidiosis than rabbits inoculated with *C. guttulatus* or *E. intestinalis* alone. In addition, we surveyed the prevalence of *C. guttulatus* in rabbits and found that the positive rate was 83% in Zhejiang Province. In summary, the results indicated that *C. guttulatus* alone is not pathogenic to healthy rabbits, although might be an opportunistic pathogen when the digestive tract is damaged by other pathogens, such as *coccidia*.

## 1. Introduction

Diarrhoea is very common in rabbits, especially in weanling rabbits, and causes huge losses in rabbit production. According to some reports, more than 50% of mortalities in rabbits may be caused by intestinal disease with diarrhoea [1,2,3]. Currently, more than twenty microorganisms, including viruses (e.g., *Lapine rotavirus*), bacteria (e.g., *Escherichia coli*, *Salmonella typhimurium*, *Clostridium welchii* and *Pasteurella multocida*) and parasites (e.g., *Eimeria spp.*, *Passalurus ambiguus*, *Cryptosporidium spp.* and *Giardia duodenalis*) have been identified as pathogens which cause diarrhoea in rabbits [4,5,6,7,8,9,10,11,12,13,14,15]. In addition to the above pathogens, *Cyniclomyces guttulatus*, a commensal yeast in the rabbit gastrointestinal tract, is also commonly seen in diarrhoea cases. However, it is unclear whether this organism causes or is a co-cause of diarrhoea with other pathogens. Some researchers believe that *C. guttulatus* is not a pathogen which causes diarrhoea, but is probably a salubrious normal inhabitant based on its common existence in healthy animals and the absence of clinical signs in experimental rabbits inoculated with *C. guttulatus* isolates [16,17]. However, other researchers believe this organism could be an opportunistic pathogen based on the large numbers of yeast cells in the faeces of diarrhoeic animals and the positive response of some diarrhoeic cases to antifungal treatment with nystatin [18,19,20,21,22]. To clearly establish the relationship between *C. guttulatus* and diarrhoea in rabbits, a *C. guttulatus* strain was isolated and identified from a New Zealand White rabbit with severe diarrhoea, and the culture conditions were optimised. Then, its relationship with rabbit diarrhoea was investigated through the inoculation of *C. guttulatus* alone and coinoculation with an intestinal protozoan, *Eimeria intestinalis*. In addition, the prevalence of *C. guttulatus* in rabbits was surveyed in Zhejiang Province, China.

## 2. Results

### 2.1. A Cyniclomyces guttulatus Zhejiang Strain Was Isolated, and The Culture Conditions Were Optimised

A *C. guttulatus* Zhejiang strain was isolated from a diarrhoeic rabbit, cultivated in YPG medium at low pH, and then identified by light microscopy. Microscopically, the vegetative cells of *C. guttulatus* were ellipsoid, colourless and approximately 20–50 μm in length and occupied two large vacuoles in the cytoplasm [Figure 1A,C]. *C. guttulatus* formed pseudohyphae when it was cultivated in a stationary culture station [Figure 1B]. When it was cultivated with rotation, free vegetative cells were clearly visible. This organism could aerobically grow at a temperature range of 36 to 42 °C and a pH range of 1.5 to 4.5 in liquid YPG medium [Figure 1D,E]. The optimal culture medium pH was 4.5 [Figure 1D]. The logarithmic growth phase was 24 to 60 h at various temperatures and a pH value of 4.5 [Figure 1E]. At a culture temperature of 40 °C and pH 4.5, the cell density of *C. guttulatus* reached 4.62 × 10^7^ ± 3.98 × 10^6^ cells/mL over 60 h from the initial culture density of 1 × 10^4^ cells/mL [Figure 1E].

### 2.2. The C. guttulatus Zhejiang Strain Showed a Close Relationship with Reference Strains Originating from Herbivores

The sequence length of the 18S fragment of the *C. guttulatus* Zhejiang strain was 1559 bp. It showed 98% sequence identity and 100% coverage with that of the *C. guttulatus* NRRL Y-17561 strain reported in GenBank (accession number JQ698886.1) [Figure 2A]. In the phylogenetic tree based on the 18S fragment, the Zhejiang strain was clustered and formed a sister clade with the NRRL Y-17561 strain. The sequence length of the 26S fragment of the strain was 632 bp and showed 100% (95% coverage), 100% (93% coverage), 97.2% (84% coverage), 96.6% (96% coverage) and 95.9% (95% coverage) identity with those of the CGDPA-GP1, DPA-CGR1, Dog-1, DPA-CGD1 and NRRL Y-17561 strains, respectively. In the 26S phylogenetic tree, this strain was positioned in the same clade as CGDPA-GP1 and DPA-CGR1 and formed a sister clade with the Dog-1, DPA-CGD1 and NRRL Y-17561 strains [Figure 2B]. According to data in GenBank, the *C. guttulatus* CGDPA-GP1 and DPA-CGR1 strains originate from herbivores, guinea pigs and rabbits, respectively, whereas the *C. guttulatus* Dog-1 and DPA-CGD1 strains originate from dogs. Thus, the *C. guttulatus* Zhejiang strain showed a closer phylogenetic relationship with the strains originating from herbivores than those from carnivores.

### 2.3. The C. guttulatus Zhejiang Strain Is Nonpathogenic to Healthy Rabbits

All SPF rabbits were healthy and were negative for *C. guttulatus* before the experiment. Two days after inoculation, *C. guttulatus* was found in rabbit faeces of all three groups (G1–G3) that were inoculated with 1 × 10^6^, 1 × 10^7^ and 10^8^
*C. guttulatus* vegetative cells per rabbit. However, no rabbits in these groups showed clinical signs of illness. Interestingly, the *C. guttulatus*-inoculated groups (G1–G3) had less feed waste than the control group (G4) [Figure 3B,C]. The mean body weight of the inoculated groups was slightly higher than that of the control group, although the difference was not statistically significant (*p* > 0.05) [Figure 3A]. Autopsy showed no macroscopic or microscopic lesions in the GI tracts of inoculated rabbits despite a large number of *C. guttulatus* cells in the gastric and intestinal contents. In particular, a thick layer of *C. guttulatus* cells colonised the gastric mucosa [Figure 3D–F]. PAS-stained gastric tissue sections showed a dense layer of saccharides on the gastric mucosa and on the cell wall of *C. guttulatus* [Figure 3G]. The *C. guttulatus* cells probably attached to the stomach mucosa through these filamentous saccharides, as shown by transmission electron microscopy [Figure 3H,I]. Our findings indicated that the *C. guttulatus* Zhejiang strain colonised the gastric mucosa, but it was nonpathogenic to healthy rabbits.

### 2.4. C. guttulatus May Be an Opportunistic Pathogen in Rabbits Infected with E. intestinalis

All SPF rabbits were healthy and were negative for *C. guttulatus* and *E. intestinalis* before the experiment. Rabbits preinoculated with *C. guttulatus* (CG/EI group) developed more severe illness and intestinal lesions following *E. intestinalis* infection than those not inoculated with *C. guttulatus* (EI group). Compared with the EI group, more severe diarrhoea, loss of appetite and constipation were observed in the CG/EI group. The number of *C. guttulatus* vegetative cells in the faeces of the CG/EI group was significantly higher than that of the CG group (*p <* 0.05) at 9 and 10 days after *E. intestinalis* infection. No *C. guttulatus* was detected in the EI and NON groups [Figure 4A]. The mean cells per gram of faeces in the CG/EI group were 1.22 × 107 ± 1.38 × 106 on day 9 and 8.55 × 106 ± 5.52 × 10^5^ on day 10, 4.7-fold and 3.2-fold higher than those of the EI group, respectively. In contrast, coccidian reproduction was lower in co-inoculated rabbits than in rabbits inoculated with *E. intestinalis* alone, as shown by a markedly lower oocyst output in the CG/EI group than in the EI group; the total faecal oocyst count per rabbit on day 10 was 2.87 × 10^9^ ± 8.13 × 10^7^ in the coinoculated group compared with 4.57 × 10^9^ ± 6.83 × 10^7^ in the group without yeast inoculation. No *E.*
*intestinalis* was detected in either the CG or NON groups [Figure 4B]. In addition, the peak oocyst excretion of the CG/EI group was also lower than that of the EI group (1.04 × 10^9^ ± 2.45 × 10^7^ on day 11 and 1.61 × 10^9^ ± 3.43 × 10^7^ on day 13). Faecal oocyst excretion in the CG/EI group peaked and cleared earlier than that in the EI group [Figure 4B]. Thus, *C. guttulatus* proliferation was enhanced by *E. intestinalis* infection, whereas *E. intestinalis* reproduction was suppressed by *C. guttulatus*. In addition, one rabbit in the CG/EI group died on the 10th day post-infection. The dead rabbit had disseminated haemorrhage and nodules in the lower jejunum and ileum, and a large number of *E. intestinalis* oocysts and *C. guttulatus* vegetative cells were detected in the small intestine [Figure 4D,E]. In addition, the mean body weight of the CG/EI group was slightly lower than that of the CG group 11 to 17 days post-infection [Figure 4C]. In summary, rabbits inoculated with both *E. intestinalis* and *C. guttulatus* developed more severe clinical signs and intestinal lesions (and one death) than rabbits inoculated with *E. intestinalis* only, associated with higher *C. guttulatus* and lower *E. intestinalis* output. The findings suggested that *C. guttulatus* might be an opportunistic pathogen in rabbits with coccidia.

### 2.5. C. guttulatus Was Highly Prevalent in Rabbits

We conducted a survey of rabbits carrying *C. guttulatus* by analysing *C. guttulatus* cells in faeces. *C. guttulatus* was detected in 210 of 253 faecal samples from rabbits over 30 days old in Zhejiang Province, and the positive rate was 83%. In this survey, the positive rate for rabbits over 60 days old was 69.7% (46/66), and for rabbits under than 60 days old, it was 87.7% (164/187). The number of rabbits with a high *C. guttulatus* load (>100 cells per microscopic field at 200× magnification) was 59 (23.3%), consisting of 4 rabbits under 60 days old and 55 rabbits over 60 days old (6.1% and 29.4% of the age group, respectively). Thus, *C. guttulatus* was highly prevalent in healthy rabbits, especially in older rabbits.

## 3. Discussion

*C. guttulatus* is a monotypic yeast genus of the Saccharomycetaceae family and inhabits the GI tract of many animal species, including rabbits, dogs and guinea pigs [17,22,23]. *C. guttulatus* was first described more than 60 years ago and is commonly found in rabbit faeces. A large number of *C. guttulatus* cells are often found in the faeces of diarrhoeic rabbits, but it is unknown whether *C. guttulatus* causes diarrhoea in rabbits. We isolated a *C. guttulatus* Zhejiang strain from a rabbit with severe diarrhoea. At optimised culture pH and temperature, a single clone was expanded and studied for its pathogenicity in rabbits. We demonstrated that the *C. guttulatus* Zhejiang strain is not a primary pathogen which causes diarrhoea in healthy SPF rabbits. An inoculation rate as high as 1×10^8^ vegetative cells per rabbit did not result in any clinical signs of illness or gastrointestinal lesions. This is consistent with previously reported studies, where rabbits orally or intravenously inoculated with *C. guttulatus* isolates showed no clinical signs of illness [16,17,24]. Unexpectedly, we found that rabbits inoculated with *C. guttulatus* showed better performance in terms of body weight gain and food intake. *C. guttulatus* seems to be a probiotic microorganism in rabbits, especially in weanling rabbits. However, the abundant vegetative cells of *C. guttulatus* commonly seen in the faeces of rabbits with diarrhoea suggest that it may be a causative microorganism of GI tract disturbance.

Some authors have proposed that *C. guttulatus* could be an opportunistic pathogen or play a co-causative role in diarrhoea of its host based on indirect evidence that anti-fungal agents, such as nystatin, are effective in the treatment of some diarrhoeic cases [19,20,21,22]. In our study, *C. guttulatus* was proven to be an opportunist through coinfection with the coccidian species *E. intestinalis*, a parasite which causes diarrhoea and intestinal lesions in rabbits. Mortality and more severe clinical signs and intestinal lesions were observed in rabbits coinfected with *C. guttulatus* and *E. intestinalis* than in rabbits inoculated with *E. intestinalis* only. Compared with rabbits inoculated with *C. guttulatus* alone, vegetative cells of *C. guttulatus* prolifically multiplied in the coinfection group, peaking at 9–10 days post-*E. intestinalis* infection. This time period is consistent with that of intestinal lesions of *E. intestinalis* infection [25,26]. The substantial multiplication of *C. guttulatus* was probably a consequence of the altered GI environment in the rabbits from *E. intestinalis* infection. Compared with rabbits with *E. intestinalis* infection alone, *E. intestinalis* oocyst excretion in coinfected rabbits decreased by 37%. We speculate that the rapid and considerable multiplication of *C. guttulatus* vegetative cells could contribute to severe symptoms in coinfected rabbits, and this multiplication of *C. guttulatus* also limited the reproduction of *E. intestinalis*.

In addition, our epidemiological survey showed that *C. guttulatus* was prevalent in rabbits in Zhejiang, China. The positive rate in rabbits was as high as 83%. This rate was significantly higher than that in other host animals, such as dogs, in which a prevalence of 14–21% has been reported [20,21,27]. Coccidia are also highly prevalent in rabbits. According to a previous study, the overall prevalence of rabbit coccidia is 41.9% in China and as high as 70% in some regions [12]. Therefore, *C. guttulatus* may contribute to the morbidity and mortality of rabbits with coccidiosis.

## 4. Materials and Methods

### 4.1. Study Design

This study was divided into two parts. In the first part, the relationship between *C. guttulatus* and rabbit diarrhoea was investigated. A *C. guttulatus* strain was first isolated and identified from a diarrhoeic rabbit with special YPG medium (pH 1.5). To investigate the pathogenicity of *C. guttulatus* alone on rabbits, sixteen SPF rabbits were randomly divided into 4 groups with 4 rabbits per group. Three groups of SPF rabbits were inoculated with 1 × 10^6^, 1 × 10^7^ or 1 × 10^8^
*C. guttulatus* vegetative cells per rabbit. One group was used as an uninoculated control. The physical state of the rabbits and possible clinical signs were observed every day post-inoculation. To observe possible pathological changes in the GI tract, all of the experimental rabbits were administered dexmedetomidine hydrochloride (5 μg/kg IM; Dexdomitor^®^, OrionPharma, Finland) and tiletamine hydrochloride-zolamide hydrochloride (0.5 mg/kg IM; Zoletil^®^50, Virbac, France) before euthanasia through air injection (20 mL/kg, IV) with sterile syringes (Haiers, Changzhou, China) in the animal diagnosis and treatment centre of our institute 18 days after inoculation according to the method of Close B. (1997) [28]. Then, the GI tracts were obtained through abdominal operation and prepared as tissue slices or wet mounts. To investigate whether *C. guttulatus* is an opportunistic pathogen, another sixteen SPF rabbits were used. They were divided into 4 groups with 4 rabbits per group. Two groups of SPF rabbits were inoculated with 1 × 10^7^
*C. guttulatus* vegetative cells per rabbit, and the other two groups were uninoculated. Fourteen days later, one of the inoculated and uninoculated groups was infected with 1×10^4^
*E. intestinalis* oocysts per rabbit. Clinical signs, oocyst output and *C. guttulatus* cells in faeces were examined post-inoculation/infection. In the second part, to survey the prevalence of *C. guttulatus* in rabbits, fresh faecal samples collected from rabbitries in Zhejiang Province, China were examined.

### 4.2. Rabbit Management

Thirty-two SPF New Zealand White rabbits (all male, 17 to 25 days old, weighing 0.2 to 1 kg) were acquired from Pizhou Dongfang Rabbit Breeding Co., Ltd. (Pizhou, China) and reared in our institute. Three days before the experiment, all rabbits were given itraconazole with free access to water by adding a dosage of 100 mg/L to avoid contamination by *C. guttulatus* during transportation, and the faeces of the experimental rabbits were examined every day under microscopy to check the existence of oocysts. They were reared within RBC6H5-independent ventilation cages (Suzhou Suhang Technology Equipment Co., Ltd., Suzhou, China at room temperature of 20 to 25 °C, with one rabbit per cage and allowed access to pathogen-free food and drinking water that was sterilised in an oven before use. Their physical condition was monitored every day during all experimental procedures. In addition, pre-weaned SPF rabbits were supplied with carrots and 0.5% milk powder in their drinking water. All animals were confirmed to be negative for *C. guttulatus* and coccidia in faeces with microscopy examination before use.

### 4.3. Isolation and Cultivation

A *C. guttulatus* Zhejiang strain was isolated from a severely diarrhoeic rabbit. Approximately 0.5 g of intestinal content was diluted to approximately 500 vegetative cells of *C. guttulatus*/mL with sterilised distilled water. Then, a 20 μL suspension was added to 10 mL YPG (pH 1.5) medium supplemented with 100 mg/L ampicillin. The strain was cultured in a 96-well culture plate with 100 μL of medium per well at 37 °C with 10% CO_2_ for 2 h. Wells with a single *C. guttulatus* vegetative cell were further cultured for 5 days before monoclonal cells of *C. guttulatus* were smeared and cultivated on solid YPG plates (pH 4.5) supplemented with 100 mg/L ampicillin at 37 °C with 10% CO_2_. A single colony of *C. guttulatus* was transferred to liquid YPG medium (pH 4.5) and cultivated at 37 °C on an orbital shaker (Zhichu, China) at a constant rotating speed of 200 rpm. Yeast multiplication was monitored using an automatic microbial growth curve analyser (Bioscreen, Helsinki, Finland) and an optical density scanner (Bug Lab, Raleigh, NC, USA). Culture conditions were optimised by varying the medium pH and culture temperature.

### 4.4. Morphological and Molecular Identification

The morphology of the *C. guttulatus* Zhejiang strain was observed under a light microscope at 400× magnification. Molecular identification was performed by PCR and gene sequencing. Two specific primer pairs for the small subunit (18S) and large subunit (26S) ribosomal RNA genes were synthesised according to Kurtzman C.P. (1998) [29]. The primer pairs were as follows: 18S upper primer (TACGGTGAAACTGCGAATGG), 18S lower primer (GCTGATGACTTGCGCTTACT), 26S upper primer (GCATATCAATAAGCGGAGGAAAAG) and 26S lower primer (GGTCCGTGTTTCAAGACGG). The PCR conditions for the 18S DNA fragment were an initial denaturation at 95 °C for 5 min, followed by 30 cycles of 94 °C denaturation for 45 s, primer annealing at 55 °C for 45 s, and extension at 72 °C for 90 s. A final primer extension of 10 min at 72 °C completed the amplification process. The amplification for 26S was the same as that for 18S, although the extension at 72 °C was extended to 120 s during the PCR cycles. *C. guttulatus* vegetative cells were directly used as the template for PCR. The PCR products were examined and separated by 1% agarose gel electrophoresis. The target bands were purified by a gel extraction kit and sequenced by Sangon Biotech (Shanghai, China). The sequenced 18S and 26S gene fragments of the *C. guttulatus* Zhejiang isolates were submitted to GenBank with the Bankit procedure and blasted in the NCBI (National Center for Biotechnology Information, Bethesda, MD, USA) database. Similar reference sequences were retrieved. Phylogenetic analysis was performed using the MegAlign program (DNAstar Inc., Madison, WI, USA). The phylogenetic tree was constructed using the Clustal W method. Information about *C. guttulatus* isolates and other yeast species used in the construction of the phylogenetic trees is listed in Table 1.

### 4.5. Inoculation of Cyniclomyces guttulatus in Rabbits and Examination of Yeast Colonisation in the GI Tract

Three groups of 20-day-old SPF rabbits (*n* = 4) were orally inoculated with 1 × 10^6^, 1 × 10^7^ or 1 × 10^8^
*C. guttulatus* vegetative cells per rabbit and designated G1, G2 and G3, respectively. Another group (G4) was not inoculated with yeast and served as the control. Body weight, activities, appetite and excreta were recorded before and after yeast inoculation. Faeces from all groups were collected daily and examined for yeast cells under a light microscope.

All rabbits were euthanised 18 days after inoculation according to the study design. The contents and the mucous layer of the stomach, duodenum, jejunum, and ileum were collected, smeared and microscopically examined. Yeast was observed in the stomach (see the Results section); therefore, tissues from different regions of the stomach were immediately frozen for microscopic examination and fixed in 10% formalin for the preparation of paraffin sections for microscopic observation after staining with PAS (periodic acid–Schiff) and in 5% glutaraldehyde for the preparation of electron microscopy sections for observation by transmission electron microscopy.

### 4.6. Coinfection of Cyniclomyces guttulatus and Eimeria intestinalis in Rabbits

Two groups (designated CG and CG/EI) of 28-day-old SPF rabbits were orally inoculated with 4 × 10^7^
*C. guttulatus* cells per rabbit (*n* = 4). After 14 days, CG/EI and another group (designated EI, without *C. guttulatus* inoculation) were infected with 1×10^4^ sporulated oocysts of *E. intestinalis*. One group (designated NON) served as the uninfected control. Body weight, activity, appetite and excreta of animals were recorded before and after inoculation. Faeces were collected for coccidial oocysts and yeast cell counting. *E. intestinalis* oocysts in faeces were counted between 10 and 16 days after infection using the McMaster method as described previously [30]. Vegetative cells of *C. guttulatus* were counted from 2 days before to 14 days after the infection of *E. intestinalis*. Briefly, 1 g of faeces was mashed with a glass stick and mixed with 60 mL of tap water. The faecal suspension was filtered through a 100-mesh sieve. *C. guttulatus* cells in the filtrate were counted in a haemocytometer under a light microscope at 100× magnification.

### 4.7. Prevalence Survey

Faecal samples were collected from 253 healthy rabbits in four regions, including Fuyang, Haining, Deqing and Wencheng (abbreviated: FY, HN, ZX and WC), in Zhejiang Province, China. The sample numbers from FY, HN, DQ and WC were 50, 50, 49 and 104, respectively. Among the surveyed rabbits, 66 were under 60 days old, and 187 were over 60 days old. The collected samples were stored at 4 °C and examined within one week. The examination was performed as follows: 2 g faeces was mixed with 60 mL of tap water. The mixture was filtered through a 100-mesh sieve. Then, 100 μL filtrate was collected for wet mount examination under a light microscope at 100× magnification. Twenty fields were observed for each sample.

### 4.8. Statistical Analysis

Statistical analyses were performed for *C. guttulatus* cell counts, rabbit body weight, and *E. intestinalis* oocyst counts by GraphPad Prism 5.01. Data are expressed as the mean ± standard deviation of four replicates, and experimental groups were compared by using paired-samples *t*-tests. Differences between groups with *p*-values < 0.05 were considered statistically significant.

## 5. Conclusions

As a commensal yeast, *C. guttulatus* is very common in rabbits and is usually not pathogenic even when the dosage is up to 1 × 10^8^ in healthy rabbits; this species seems to be a probiotic microorganism in rabbits, but it may be an opportunistic pathogen when the GI environment is altered by enteric pathogens such as coccidia. Considering the high prevalence of both *C. guttulatus* and coccidia in rabbits, the potential harm of *C. guttulatus* to the rabbit industry warrants attention and needs further study.

## Figures and Tables

**Figure 1 pathogens-10-00880-f001:**
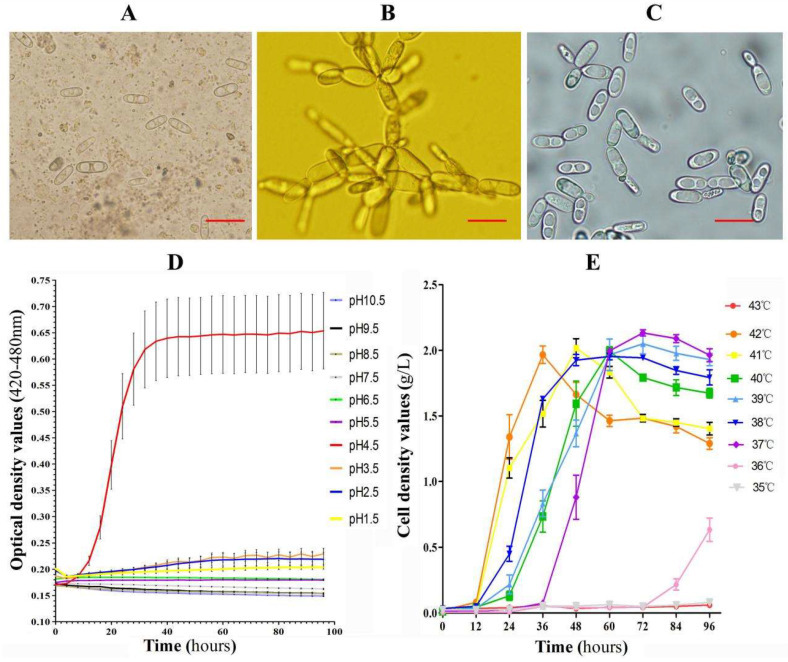
*Cyniclomyces guttulatus* was isolated from a rabbit with diarrhoea, and culture conditions were optimised. (**A**) *C. guttulatus* vegetative cells in the faeces of a diarrhoeic rabbit. (**B**) Pseudohyphae formed when *C. guttulatus* was cultivated under stationary culture conditions. (**C**) Free vegetative cells formed when *C. guttulatus* was cultivated with rotation. (**D**) Growth curves of *C. guttulatus* at different pH values. (**E**) Growth curves of *C. guttulatus* at different incubation temperatures. Bar: 20 μm.

**Figure 2 pathogens-10-00880-f002:**
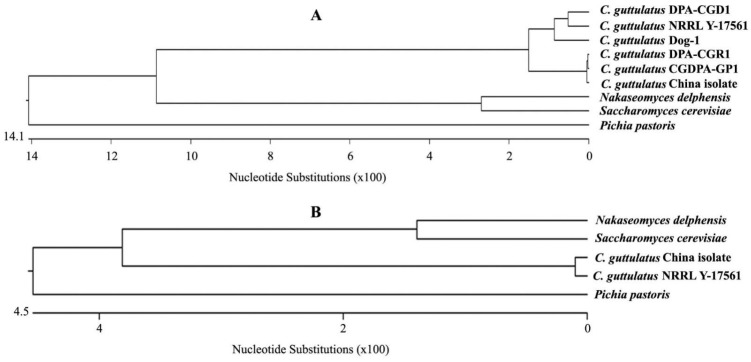
Phylogenetic analysis of the *Cyniclomyces guttulatus* Zhejiang isolate based on 26S and 18S ribosomal RNA gene sequences. (**A**) Phylogenetic tree of the *C. guttulatus* Zhejiang isolate and 5 other isolates of *C. guttulatus* and 3 other yeast species based on the 26S ribosomal gene sequence. (**B**) Phylogenetic tree of the *C. guttulatus* Zhejiang isolate, one isolate of *C. guttulatus* and 3 other yeast species based on the 18S ribosomal gene sequence.

**Figure 3 pathogens-10-00880-f003:**
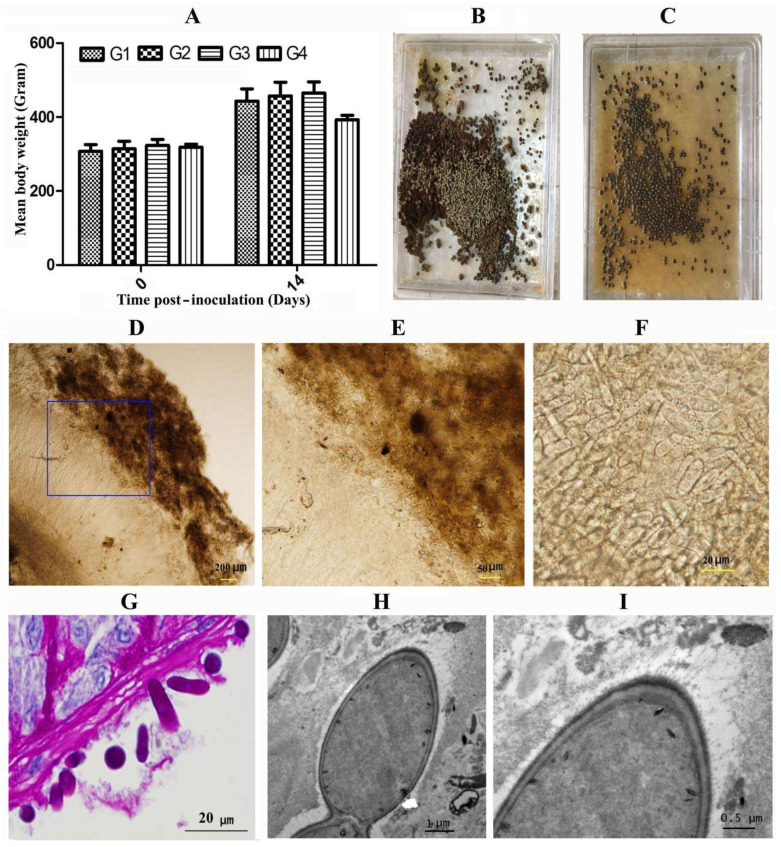
Effects of the *Cyniclomyces guttulatus* Zhejiang isolate in healthy SPF rabbits. (**A**) Mean body weight of rabbits inoculated with 1 × 10^6^ (G1), 1 × 10^7^ (G2) and 1 × 10^8^ (G3) *C. guttulatus* vegetative cells per rabbit and the control group (G4). (**B**,**C**) Excreta trays for rabbits uninoculated and inoculated with *C. guttulatus*, respectively, showing unconsumed feed in the tray of the uninoculated group. (**D**,**E**) Microscopic observation of a frozen cross-section of gastric pylorus from a rabbit inoculated with 1 × 10^8^
*C. guttulatus* at 40 and 100× magnification. (**F**) Microscopic observation of the gastric fundic mucosa layer from a rabbit inoculated with *C. guttulatus* at 200× magnification. (**G**) Microscopic observation of a paraffin section of the gastric fundus from a rabbit inoculated with 1 × 10^8^
*C. guttulatus* with PAS staining. (**H**,**I**) Transmission electron microscopic observation of the gastric fundus from a rabbit inoculated with 1 × 10^8^
*C. guttulatus*.

**Figure 4 pathogens-10-00880-f004:**
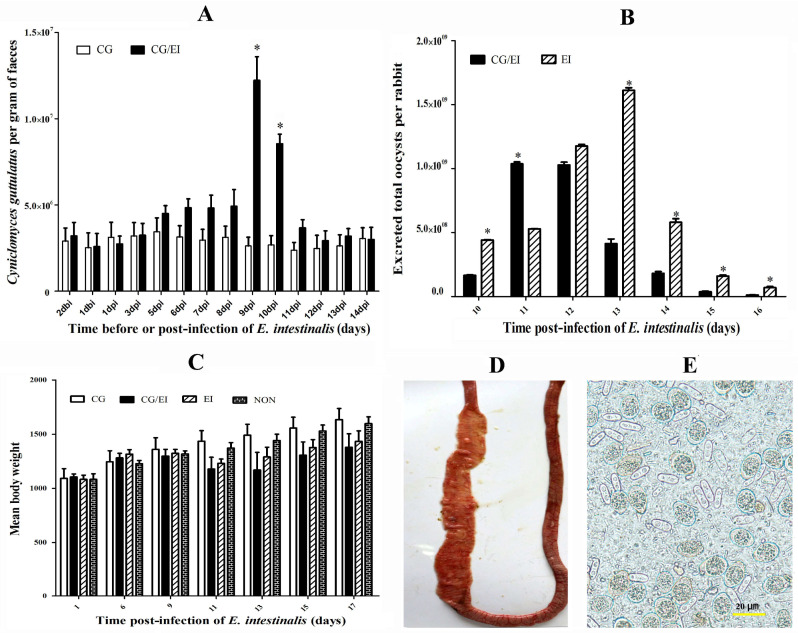
Effects of co-inoculation of SPF rabbits with *C. guttulatus* and *E. intestinalis*. (**A**) Faecal excretion of *C. guttulatus* in rabbits inoculated with *C. guttulatus* only (CG) or inoculated with *C. guttulatus* and *E. intestinalis* (CG/EI). dbi/dpi= days before the infection of *E. intestinalis*/days after the infection of *E. intestinalis*. (**B**) Faecal oocyst counts in rabbits 10 to 16 days after infection with *E. intestinalis* without preinoculation with *C. guttulatus* (EI) or inoculation with both *C. guttulatus* and *E. intestinalis* (CG/EI). (**C**) Body weight of rabbits inoculated with *C. guttulatus* only (CG), co-inoculated with *C. guttulatus* and *E. intestinalis* (CG/EI), or with *E. intestinalis* only (EI), and control rabbits (NON) (**D**,**E**) Gross lesions of the small intestine and *C. guttulatus* vegetative cells and *E. intestinalis* oocysts in faeces of the dead rabbit co-inoculated with *C. guttulatus* and *E. intestinalis*. “*” indicated significant difference of datas between CG/EI and CG or EI groups (*p* ≤ 0.05).

**Table 1 pathogens-10-00880-t001:** GenBank sequences used in the construction of the phylogenetic tree.

Species/Isolate	Location/Source	Genbank Accession No.
*Cyniclomyces guttulatus* Zhejiang strain	Intestinal content of rabbit in China	MN633294/ MN625917
*Cyniclomyces guttulatus* isolate DPA-CGR1	Rabbit faeces in Brazil	JQ861267.1
*Cyniclomyces guttulatus* isolate DPA-CGD1	Dog faeces in Brazil	JQ861266.1
*Cyniclomyces guttulatus* isolate NRRL Y -17561	-	U76196.1/ JQ698886.1
*Cyniclomyces guttulatus* isolate CGDPA-GP1	Guinea pig in Brazil	KC484339.1
*Cyniclomyces guttulatus* dog-1	Dog faeces in Norway	FJ755179.1
*Nakaseomyces delphensis*	-	JQ689014.1
*Saccharomyces cerevisiae*	-	JQ689017.1
*Pichia pastoris*	-	U75963.1

Note: “-” indicates no information detailed.

## Data Availability

Data supporting the conclusions of this article are included within the article and the additional files. The newly generated sequences were submitted to the GenBank database under the accession numbers MN633294 and MN625917.

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
