# Peer review of "An Investigation of the Relationship between Cyniclomyces guttulatus and Rabbit Diarrhoea"

_pathogens, 2021, doi:10.3390/pathogens10070880_

Round 1

Reviewer 1 Report

In this manuscript entitled: “An investigation of the relationship between Cyniclomyces guttulatus and rabbit diarrhea” the authors aimed to determine the pathogenicity of C. guttulatus in rabbits. The data presented in this manuscript is significant and will add to the literature for this potential pathogen. I respectfully have a few recommendations that will elevate the data in this paper for consideration. Additionally, it is evident that this manuscript is not written by a native English speaker and some edits in sentence structure are required prior to publication; examples lines 101, 128.

Major Revisions Needed:

  1. One of the major differences between the groups in the co-infection model is intestinal lesions, however these were no presented in a grading scheme. The authors should revise this manuscript to include this information (this should be quantified and statistical analysis should be performed).

  1. Abstract: This needs to be rewritten, the sentence structure and purpose of the paper is not as clear as the introduction is written. Please revise.

  1. Figure 4: The patterns in the bar graphs are distracting to the reader. Consider changing. Example: White and black. Any values that are significant different should be denoted in the figure.

  • Figure 4 A: Did you check for C guttulatus in the control groups (EI and NON), this should be included in the results?
  • Figure 4B, Did you check for E. intestinalis in the control groups (CG and NON), this should be included in the results?
  • Figure 4C, there does not appear to be a different between the CG/EI and EI groups, in regards to body weight. Is this correct?

  1. Figure 1 D: It is hard to see the lines due to the size of the data points. Please revise.

  1. Line 245: How was it determined that rabbits were SPF? Was it microscopy or PCR for the C. guttulatus and E. intestinalis testing? I would be concerned to say that these animals had no C.guttulatus with a single fecal sample examined by microscopy. This is a flaw of this paper. The authors should clarify.

Minor edits:

Line 105: Define what feed waste is. Do you mean less fecal volume? But in Figure 3 B/C it appears to be related to feed that was left behind? Please clarify.

Line 238: F in feces should not be capitalized

Reviewer 2 Report

The research has adequately described the pathogenicity of C. guttulatus in rabbits and the authors concluded that  C. quttulatus not pathogenic to healthy rabbits but may be an opportunistic pathogen when the digestive tract is damaged by other pathogens such as coccidia. The Ms contains useful information and help full to rabbits breeders.  I have several comments that need to be addressed in the text related to 

  1. The novelty of the study
  2. From of feed used.
  3. The details of the statistical analyses such as the experimental unit, the statistical model used, and name of test used for mean differences 
  4. Comments about the maximum tolerated dose of use C. quttulatus in rabbits.

Round 2

Reviewer 1 Report

Thank you for addressing the previous comments. Please see a few additional comments below.

Lines 243-244: Point of clarification: The authors need to provide more information about the use of itraconazole in their study. Were all animals on this antifungal? What duration? Were animals infected with C. guttulatus also receiving this anti-fungal? These details are important for using this animal model and allowing the reader to replicate the experiments if they choose.

Abstract: Lines 20-21, 32-33 that were added need to be edited by a native English speaker.

Line 133. “no C. guttulatus was 133 detected in EI and NON groups”; this should be its own sentence.

Line 140: “no E. intestinalis was detected in 140 both CG and NON groups”; this should be its own sentence.

Author Response

Lines 243-244: Point of clarification: The authors need to provide more information about the use of itraconazole in their study. Were all animals on this antifungal? What duration? Were animals infected with C. guttulatus also receiving this anti-fungal? These details are important for using this animal model and allowing the reader to replicate the experiments if they choose.

Response:Thanks for your construction suggestion. We have added this content in our revised manuscript, line 249-251, page 9.

Abstract: Lines 20-21, 32-33 that were added need to be edited by a native English speaker.

Response:Thanks for your careful review. We have revised abstract part in our revision.

Line 133. “no C. guttulatus was 133 detected in EI and NON groups”; this should be its own sentence.

Line 140: “no E. intestinalis was detected in 140 both CG and NON groups”; this should be its own sentence.

Response:Thanks for your careful review. We have revised these two parts in line 137 and line 144 in our revision.